# Walking Engagement in Mexican Americans Who Participated in a Community-Wide Step Challenge in El Paso, TX

**DOI:** 10.3390/ijerph182312738

**Published:** 2021-12-02

**Authors:** Stefan Saadiq, Roy Valenzuela, Jing Wang, Zenong Yin, Deborah Parra-Medina, Jennifer Gay, Jennifer J. Salinas

**Affiliations:** 1Graduate School of Biomedical Sciences, Texas Tech University Health Sciences Center, El Paso, TX 79905, USA; ssaadiq@ttuhsc.edu; 2Department of Molecular and Translational Medicine, Paul L. Foster School of Medicine, Texas Tech University Health Sciences Center, El Paso, TX 79905, USA; royvalen@ttuhsc.edu; 3College of Nursing, Florida State University, Tallahassee, FL 32306, USA; jingwang@nursing.fsu.edu; 4Department of Public Health, University of Texas at San Antonio, San Antonio, TX 78229, USA; Zenong.Yin@utsa.edu; 5Department of Mexican American & Latina/o Studies, Latino Research Institute, University of Texas at Austin, Austin, TX 78712, USA; parramedina@austin.utexas.edu; 6Institute of Gerontology, College of Public Health, University of Georgia, Athens, GA 30602, USA; jlgay@uga.edu

**Keywords:** obesity, Hispanics, walking, physical activity, socioeconomic inequities, El Paso, Texas, Texas–Mexico border

## Abstract

In the United States, the Latinx population has the highest prevalence of physical inactivity compared with other ethnicities. Research shows that work-based physical activity interventions have been widely implemented in the non-Latinx population and effectively increase physical activity in the non-Latinx population. In an effort to improve physical activity and reduce obesity among the Latinx population, we conducted 10,000 Steps for 100 Days, an employer-based walking challenge campaign, to increase walking engagement among Latinx employees located in El Paso, Texas. Participants reported their number of steps using a pedometer or smartphone. Step counts were collected at baseline, 2 weeks post challenge, and 6 months post challenge. Screenshots of the tracking device were uploaded to an online tracker. Regression analysis was conducted to identify covariates associated with baseline and 2-week and 6-month average daily steps. Generalized estimating equations (GEE) were performed to predict steps over time by demographic characteristics. Participation in the 10,000 Steps for 100 Days walking challenge was associated with a sustained increase in average daily steps. Participants with less than 7000 steps per day demonstrated the greatest increase in average daily steps (921 steps at 2 weeks; 1002.4 steps at 6 months). Demographic characteristics were not significant predictors of average steps, except that married participants had higher average steps. Participants with 10,000 or more daily steps had a 51% (*p* = 0.031) higher chance of having a professional occupation than a non-professional one compared to those with 7000 or fewer daily steps. We provided initial evidence that the walking challenge is an effective approach for improving physical activity in the Latinx population.

## 1. Introduction

Obesity is a major risk factor for preventable cardiometabolic diseases, cancer, and premature death (cardiovascular and metabolic consequence of obesity) [1,2,3,4,5]. The Latinx population is disproportionally represented in the U.S. among those who are considered obese (44.8%), second only to Black Americans [6]. Contributing to this disparity is low engagement in physical activity and excess sedentary time [7,8]. Nationally, the Latinx population has the highest prevalence of physical inactivity at 31.7% compared to the national average (15%) [7]. This is a major public health problem as the Latinx population is disproportionately impacted by diseases that could be modified or prevented with adequate physical activity [5].

In an effort to improve physical activity and reduce obesity among the Latinx population, there have been numerous studies testing various approaches to increasing regular engagement [9,10,11,12,13]. While many show promise, most have focused on children and families, not taking into consideration that most Latinx individuals work one or more jobs [14], leaving approaches to increased activity during work hours largely untested in this population. However, work-based physical activity interventions have been widely tested in the non-Latinx population and demonstrate the promise for increasing physical activity and reducing obesity [15,16,17,18,19,20,21,22,23,24,25,26,27,28]. Many promote physical activity as the most promising intervention to reduce obesity [29,30,31]. Others have promoted walking through challenges and competitions [30,31,32,33]. Given the evidence base of work-based physical activity intervention, coupled with high workforce participation, using this approach may fill a major gap in knowledge in the Latinx population [34].

Walking at least 7000 steps a day has been documented to improve cardiometabolic function and reduce mortality [28,35]. Walking challenges provide an alternative way for adults to obtain sufficient levels of physical activity throughout the day, helping them reach the threshold and allowing for friendly reinforcement with teams and goal setting [35,36,37,38]. Although the amount varies, findings from these interventions have demonstrated that participants do increase daily step averages and are more likely to maintain regular walking through goal implementation [39,40,41]. However, most walking challenge programs were organized in high income countries or in urban centers with a non-Latinx population [39,40]. Therefore, in this current study, we evaluate the effectiveness of 10,000 Steps for 100 Days, an employer-based walking challenge campaign, in helping Latinx working adults who reside in a predominantly Mexican-American metropolitan area (El Paso, TX, USA) to increase their physical activity engagement.

The 10,000 Steps for 100 Days challenge was a community-wide walking challenge campaign intended to increase walking engagement among employees located in El Paso, Texas. El Paso is the second-largest majority-Mexican-American city in the United States (82.9%) [42]. Obesity in El Paso affects 37.8% of its residences, due in part to low physical activity engagement [43]. Recent Behavioral Risk Factor Surveillance Survey (BRFSS) data suggest that in El Paso, 74.6% achieve any physical activity, meaning that 25.4% of El Paso adults are sedentary [44]. This walking challenge campaign was intended to address these disparities by offering a free physical activity challenge that was accessible to most and did not require expensive equipment or memberships. The challenge program was offered to employers as a complement to potentially existent wellness or as a means for engaging employees who may be employed in sedentary occupations [45]. The challenge has been well received, as close to 1500 employees have taken part in the challenge since 2019. Furthermore, border cities such as El Paso are unique in that they have a homogenous population, allowing for the study of behavior change in the context of a large Latinx community [42].

This report provides findings from our evaluation study intended to assess walking behavioral changes in response to challenge participation. We recruited approximately 20% of participants to take part in a repeated measures study that assessed step averages at baseline and 2 weeks and 6 months post challenge completion. The purpose of this paper is to present findings on identified covariates associated with baseline and 2-week and 6-month step averages, as well as change over time, that will serve as the basis of further assessment of the effectiveness of walking challenges in improving physical activity engagement among the Latinx population.

## 2. Materials and Methods

### 2.1. Description of 10,000 Steps for 100 Days Challenge

In the 10,000 Steps for 100 Days challenge, participants were encouraged to sustain 10,000 steps daily consecutively for 100 days. Participants reported their steps on a weekly basis for a total of 14 weeks. Screenshots of daily steps were submitted to our Redcap database.

### 2.2. Participants and Eligibility

A total of 208 participants that were recruited into the evaluation study provided data at baseline. At walking challenge registration, participants were invited to take part in a voluntary prospective online evaluation study to determine the effectiveness of the challenge in improving and sustaining walking engagement after completion. All participating employees were recruited from the local city government, school districts, and local businesses. All businesses were located in El Paso County, Texas. Participants needed to be at least 18 years of age and a participant in the walking challenge. Pregnant women were excluded from the study due to physical constraints. The 208 participants who agreed to complete the online survey represented approximately 20% of overall walking challenge participants.

### 2.3. Data Collection and Measurement

At baseline, participants completed a survey and uploaded a screenshot from their tracking device from the previous week. At 2 weeks and 6 months, participants received emails with an invitation to upload a screenshot of their most recent week of steps.

#### 2.3.1. Measurement

##### Primary Outcome

Step Counts: Participants reported steps tracked using a wrist-based pedometer or smartphone. During the study, number of steps was self-reported and collected at baseline, 2 weeks, and 6 months using a wrist-based pedometer or step recording smartphone. Accuracy of both wrist-based pedometers and step recording smartphones have been validated previously [46,47,48,49,50,51]. Step total for each time point was used to calculate means and percent change over the course of the follow-up period. Step counts were also used as time point outcomes in the regression analysis. Baseline step counts were also categorized at (1) less than 7000, (2) 7000 to 9999, and (3) 10,000 or greater and used in descriptive and regression analysis to assess change by thresholds for health benefit [44,52].

##### Previous Challenge Participation

In addition to the 10,000 Steps for 100 Days, the program offers two other step challenges at different times of the year. Previous challenge participation was measured as yes/no to whether participants participated in any previous walking challenges sponsored by this program.

##### Demographic Covariates

Sociodemographic information was recorded during the baseline survey only. For this study, we controlled for age in years, U.S. born (yes/no), married (yes/no), years of education, and occupation (professional vs. non-professional). Professional occupations were considered to be professional or technical, managerial, clerical, or sales worker. Non-professional was considered to be craftsman, skilled manual worker, semi-skilled operator, service worker, or laborer. We collapsed traditional categories into two categories to capture professions that may be sedentary versus non-sedentary. With this distinction we were attempting to distinguish sedentary jobs from those requiring physical activity to perform the major requirements of the position.

### 2.4. Statistical Analysis

Frequencies, cross tabulations, and means were first conducted to assess for regularity in the data, outliers, and other inconsistencies that may affect the analysis. Categories of some variables were collapsed into other categories in cases where there were small numbers or where it made analytical sense to collapse (i.e., professional vs. non-professional). Regression analysis (OLS) was then conducted to predict by key demographic characteristics 2-week and 6-month step averages for the full sample and stratified by occupational status. Regression analysis was conducted using intent to treat principle, whereas missing step data were replaced with baseline values at 2 weeks and baseline or 2-week values at 6 months. Logistic regression analysis was conducted to determine the odds of baseline step threshold categories based on demographic characteristics and past challenge participation. Generalized estimating equations (GEE) were performed to model steps over time by demographic characteristics. All analysis was conducted using intent to treat principle, whereas missing step data were replaced with baseline values at 2 weeks and baseline or 2-week values at 6 months. Regression analysis was conducted using jackknife variance estimate that systematically removes observations, repeating analyses to minimize bias within non-parametric samples. All the statistical analyses were done by using STATA 16 SE statistical software (StataCorp LLC, College Station, TX 77845 USA). R-squared value was reported in the table to compare the effect size of independent variables. *p* < 0.05 was considered statistically significant.

### 2.5. Results

Table 1 presents participant demographic characteristics, previous walking challenge history, and baseline step threshold distribution for the 10,000 Steps for 100 Days challenge. On average, participants were 41.3 years of age, were U.S. born (83.2%), had about 3 years of college, were in professional occupations (81.2%), and had not participated in a previous challenge. They were about equally married as not married and varied in where they stood at base in terms of meeting recommended levels of steps per day. Most participants on average at baseline were at less than 7000 steps (37.0%) or between 7000 and 9999 (37.5%) per day.

Table 1 also presents average steps for each demographic category and percent differences from baseline to 2 weeks and 6 months post challenge completion. Beginning with the overall averages, at baseline, participants walked on average 8202 steps, increasing by 1.4% to 8320 at the end of the challenge. At 6 months, participants increased their average steps to 8435 per day, which is 2.8% higher than at baseline. Changes from baseline varied by demographic characteristic. U.S.-born participants and immigrants shared similar increases in steps at 2 weeks, but at 6 months, immigrants’ improvement was double that of the U.S.-born participants (5.2% immigrants vs. 2.3% U.S. born). Married participants increased their steps by 4.3% at 2 weeks post challenge and increased that percentage slightly to 4.9% after 6 months. However, those not married decreased at 2 weeks by 1.7%, and at 6 months, they were less than 1% (0.6%) greater than at baseline. Participants in professional occupations and non-professionals had similar percentage increases at 2 weeks (6.1 and 5.9%, respectively); however, by 6 months, the increase had been reduced to 2.2% for professionals but increased slightly for the non-professionals to 6.2%.

We conducted an adjusted regression analysis predicting 2-week and 6-month challenge completion steps in order to identify key demographic covariates, assess the effect of baseline step counts, and stratify by occupation to assess variation between professional and non-professional employees (see Table 2). Overall, demographic characteristics were not important predictors of step averages, with the exception of marital status. At 2 weeks post intervention, being married contributed to an average of 725.0 (*p* = 0.027) steps more than being unmarried. However, in the stratified by occupational status analysis, it appears that the married advantage is only observed among professional employees 776.8 (*p* = 0.033) professional vs. 129.3 (*p* = 0.858) non-professional). At 6 months, the married advantage is no longer significant overall (605.3, *p* = 0.114) but near significant in professional participants (790.3, *p* = 0.070).

It is not surprising that baseline step averages were a highly significant predictor of 2-week and 6-month post challenge step averages. The model accounted for more than 60% of the variance explained and more than 70% in non-professionals. In a separate analysis, not shown, the variance explained by demographic characteristics alone was approximately 3%. Moreover, the baseline steps increase corresponded to close to a one step increase at 2 weeks and at 6 months. Therefore, the majority of the variance in 2-week and 6-month post challenge participation average steps is explained by baseline engagement. This effect, however, declined slightly at 6 months, particularly for professional participants.

In order to determine if participants in each baseline step category differ significantly demographically or by past challenge participation, a multinomial logistic regression was conducted predicting odds for 7000 to 9999 and 10,000+ to less than 7000. The only significant category was occupation for the 10,000 category (see Table 3). Participants in the 10,000 or more category had a 4.53 (*p* = 0.031) higher chance of having a professional occupation than a non-professional one compared to those in the 7000 or less category. Similarly, although only near significance, participants in the 7000 to 9999 category were 3.35 times more likely (*p* = 0.070) to be employed in a professional occupation than participants in the fewer than 7000 steps category.

Table 4 presents GEE regression results for analysis modeling step averages over time using demographic characteristics. While most demographic and occupational characteristics were not significant, marital status was marginally significant. On average, participants who were married, after adjusting for other covariates, increased the steps by 524.8 (*p* = 0.056) over the follow-up period.

## 3. Discussion

The Latinx population is over-represented among those that are obese and lack adequate physical activity [9,10,11,12,13]. In this study we evaluated a community-wide employer-based walking challenge in El Paso, Texas, a predominantly Mexican-American Latinx metropolitan area [42]. Findings from this study indicated that participation in the 10,000 Steps for 100 Days walking challenge was associated with a sustained increase in average daily steps over a 6-month follow-up period. The greatest increase was observed among those who averaged fewer than 7000 steps per day. Participants who achieved more than 7000 steps per day at baseline tended to work in professional occupations and were less likely to sustain their average daily step counts during the follow-up period. Finally, the only demographic characteristic that appeared to be associated with follow-up period step averages was being married, but only for professionals.

The 10,000 Step for 100 Days walking challenge was intended to provide an opportunity to be active in a population with low physical activity engagement [44]. The guiding principle for the design of this study was the established 7000 steps per day threshold for health benefits and the general standard of 10,000 commonly used with most industry activity trackers [28,35]. While participants who were not engaging in 7000 steps at the outset of the challenge did not, on average, reach that milestone, they did substantially improve their steps and maintained this for at least 6 months. Step challenges have been tested in a number of different contexts, including community, school, and employer settings [29,53,54]. In general, these challenges have led to meaningful weight loss, improvements in mental health, a reduction in sedentary time, and increased likelihood of meeting recommended daily physical activity levels [27,54,55,56]. Although the majority of studies have not included Latinx groups, there is a burgeoning body of literature suggesting that walking challenges are effective in increasing physical activity engagement and a promising approach for improving physical activity engagement in this population [57]. An unexpected finding was that professionals, who were expected to be more sedentary, had, on average, higher average baseline steps, were more likely to meet recommended step levels, and were less likely to increase their average steps at 2 weeks and 6 months after challenge completion, since non-professional jobs generally require more steps to perform the basic function of the position [58,59,60,61,62].The most likely explanation is that professional occupations are more likely to engage in leisure-time activities that may increase step activity [63].

Marital status was one of the only significant demographic characteristics to significantly predict steps at 2 weeks and near significantly predict steps at 6 months for professionals only. There is consistent evidence that marital status is beneficial to health in a number of ways [64,65,66,67]. The married tend to engage more in health-enhancing activities and receive regular preventive care, and there is some evidence that marriage may have a positive impact on mental wellbeing [68]. Moreover, men and women who are married tend to also be more active if their partner is also physically active [67]. Participants of the 10,000 Steps for 100 Days challenge are encouraged to sign up with spouses, significant others, friends, and family. It may be that professionals are more likely to compete with their spouse, thereby encouraging each other to keep up step averages after the challenge is completed [69]. Alternatively, more active couples may be more likely to participate in any physical activity, as it may already be something that they engage in [69]. While this study extends the knowledge of the marital benefit on physical activity to the Mexican-American Latinx population, more research is needed to establish what role marital status may play in improving physical activity engagement in the population that is sedentary.

### Strengths and Limitations

While this evaluation does provide promising evidence to support walking challenges in the Latinx population to increase walking engagement and potentially reduce the burden of cardiometabolic diseases in this high-risk group, there are notable limitations. First, this study was conducted in a location that is predominantly Mexican-American Latinx, and therefore similar studies would need to be conducted in other geographic areas with other Latinx populations before conclusions could be made across Latinx groups. Additionally, our evaluation study sample was not collected at random and is small since participants were recruited voluntarily at the time of the walking challenge registration. In an effort to address potential bias in our analysis, we conducted jackknife variance estimates; however, further analysis that includes a larger sample size is needed to confirm the effectiveness of this challenge in increasing step counts in this predominantly Mexican-American Latinx community. Additionally, we did not collect information on walking patterns that could have helped us understand our consistent finding on marital status and help tease out the effect of professional versus non-professional participants. The types of steps in the physical activity domains (occupational, domestic transportation, and leisure time) were unable to be differentiated due to limitations of the collection devices. It is unclear to what extent spouses or coworkers influence steps and maintenance of steps over the course of the follow-up, and this is important to disentangle in future analyses of these walking challenges. Finally, participants were only followed 6 months after walking challenge participation, and some of the literature suggests that the ideal follow-up window is 12 to 18 months after completion. Therefore, further studies should incorporate longer follow-up periods to assess long-term behavioral changes. Additionally, further investigation is needed to understand the impact of multiple-challenge participation, as we observed little to no effect of our challenge on these participants’ average steps. Despite the noted limitations of this study, we provide initial evidence that walking challenges may be an effective approach to improving physical activity in the Latinx population with low engagement.

## 4. Conclusions

Walking challenges may be effective at increasing physical activity engagement in Mexican Americans and other similar groups living in resource scarce areas that also have low physical activity engagement.

## Figures and Tables

**Table 1 ijerph-18-12738-t001:** Participant demographic characteristics, time point average steps, and percent differences from baseline.

		Average Steps and % Differences from Baseline				
	Participant Characteristics	Pre Challenge *	2-Week Post *	2-Week Post % Diff Baseline	6-Month Post *	6-Month Post % Diff Baseline
**Age (years) (mean + s.d.)**	41.3 (11.2)	8202.3	8320.4	1.4	8435.9	2.8
**US Born (*n* (%))**						
Yes	173 (83.2)	8144	8263.4	1.5	8335.3	2.3
No	35 (16.8)	8490.8	8601.9	1.3	8933.1	5.2
**Married (*n* (%))**						
Yes	106 (51.0)	8409.9	8773.6	4.3	8822.6	4.9
No	102 (49.0)	7986.6	7849.4	−1.7	8033.9	0.6
**Education (years) (mean + s.d.)**	15.0 (1.4)					
**Occupation**						
Professional	169 (81.2)	8423.1	8474.9	6.1	8606.3	2.2
Non-Professional	39 (18.8)	7245.4	7650.7	5.9	7697.3	6.2
**Previous challenge participation**						
Yes	32 (15.4)	8991.6	8840.5	–1.9	8950.1	−0.46
No	176 (84.6)	8058.8	8225.8	2.1	8342.4	3.5
**Recommended Step Thresholds**						
Less than 7000	77 (37.0)	4710.4	5631.3	19.6	5712.8	21.3
7000−9999	78 (37.5)	8384.4	8678.7	3.5	8952.6	6.8
10,000 +	53 (25.5)	13,007.6	11,699.9	−10	11,631.5	–10.58

* Significance level at 0.05.

**Table 2 ijerph-18-12738-t002:** Adjusted jackknife variance estimated regression results for 2-week and 6-month step counts by occupation status (professional vs. non-professional).

	2 Weeks Post Challenge			6 Months Post Challenge		
	Full β (*p* Value)	Professional β (*p* Value)	Non-Professional β (*p* Value)	Full β (*p* Value)	Professional β (*p* Value)	Non-Professional β (*p* Value)
*n*	208	169	39	208	169	39
R^2^	0.614	0.591	0.779	0.527	0.485	0.759
Number of Previous Challenges	8.68 (0.973)	−117.6 (0.611)	1958.6 (0.194)	−42.5 (0.909)	−113.5 (0.755)	1696.6 (0.375)
US Born	−124.5 (0.744)	−172.4 (0.680)	−382.0 (0.510)	−379.3 (0.355)	−477.4 (0.292)	−155.2 (0.825)
Married	725.0 **(0.027)**	776.8 **(0.033)**	129.3 (0.858)	605.3 (0.114)	790.3 (0.070)	−512.9 (0.564)
Educational Attainment	−139.1 (0.151)	−105.5 (0.326)	−123.9 (0.499)	−185.6 (0.079)	−172.8 (0.155)	−173.4 (0.369)
Age	−10.7 (0.457)	6.49 (0.681)	−39.4 (0.190)	−13.5 (0.434)	−2.03 (0.919)	−19.3 (0.582)
Baseline Steps	0.682 **(0.000)** *	0.637 **(0.000)** *	0.870 **(0.000)** *	0.653 **(0.000)** *	0.602 **(0.000)** *	0.876 **(0.000)** *
Occupation	−3.74 (0.992)			152.4 (0.706)		
Constant	4996.3 (0.011)	4179.4 (0.048)	4615.2 (0.202)	6323.4 (0.003)	6228.9 (0.007)	4636.5 (0.206)

* Significance level at 0.05. **Bold** denote significant *p*-values.

**Table 3 ijerph-18-12738-t003:** Odds ratios from logistic regression prediction baseline step thresholds adjusting for demographic characteristics (< 7000 is the reference category).

	7000–9999 (Mean + s.d.)	10,000+ (Mean + s.d.)
Age	0.966 (0.214)	1.01 (0.755)
US Born (*n* (%)) (yes = 1)	0.980 (0.978)	1.84 (0.435)
Married (*n* (%)) (yes = 1)	1.05 (0.924)	2.23 (0.168)
Education (years)	1.24 (0.280)	1.02 (0.905)
Occupation (Professional = 1)	3.35 (0.070)	4.53 (0.031)
Previous challenge participation (yes = 1)	2.36 (0.165)	2.72 (0.106)

**Table 4 ijerph-18-12738-t004:** GEE regression results predicting percent change from baseline.

	β	Std. Err.	*p* > z	95% C.I.
Age	−12.07	12.5	0.335	−36.6, 12.5
US Born (yes = 1)	−294.6	357.2	0.41	−994.8, 405.5
Married (yes = 1)	524.8	274.7	0.056	−13.6, 1063.3
Occupation (Professional = 1)	−10.7	359.3	0.976	−714.9, 693.4
Previous Challenge Participant (yes = 1)	−106.7	331.7	0.748	−756.9, 543.5
Constant	2892.0	695.3	0.000	1529.2, 4254.8

## Data Availability

Not applicable.

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
