# Peer review of "Walking Engagement in Mexican Americans Who Participated in a Community-Wide Step Challenge in El Paso, TX"

_ijerph, 2021, doi:10.3390/ijerph182312738_

Round 1
Reviewer 1 Report
Thank you for the opportunity to review the manuscript entitled "Walking Engagement in Mexican Americans who Participated in a Community-wide Step Challenge in El Paso, TX".
This study investigated the effects of an employer-based walking challenge campaign on the number of objectively measured daily steps in Latinx participants. The study provides interesting findings on this type of challenge to increase the physical activity level of the working class. The results in table 2 are excellent!
I have some notes to be clarified by the authors in the manuscript:
- The manuscript has 68 references in total, with 52 references used in the introduction section. This is too much. For example, the manuscript's first sentence uses seven references to indicate that "Obesity is a major risk factor for preventable cardiometabolic diseases, cancer, and premature death (cardiovascular and metabolic consequence of obesity)."
- I suggest the authors explain to the readers the use of the term Latinx in the introduction section.
- Reference 52 (topic 2.1) – is not related to the methodology details of this challenge. Please, revise the reference section;
- Method: Was the selection of participants by convenience? Participants were recruited from which area and companies? Please, describe in more detail the study selection process.
- My main concern is related to the use of different devices to make objective measurements of the number of steps. There is great variability between different brands. Has the smartphone step recording app been validated? Were there standardized instructions for using the different devices? How did the authors deal with this problem?
- There was a diagramming problem between table 3 and the beginning of the discussion session, and I could not read it.
- Objective measurement devices do not differentiate the number of steps in different physical activity domains (occupational, domestic, transportation, and leisure time). Do the authors consider that this may be a study limitation to understanding the factors that affect physical activity level? Or be considered a confounder?
Reviewer 2 Report
This manuscript aims to identify covariates associated with baseline, 2-week, and 6-month step count averages, as well as assess change in step count averages over time following participation in a workplace walking challenge. This manuscript targets an important and understudied population (Latinx), as well as studies an appropriate work-based physical activity program which has been largely untested within the Latinx population. This manuscript contributes early evidence for work-based walking challenges as an effective approach for increasing physical activity participation within the Latinx population.
General Concept Comments
The authors have done a great job defining the “problem” and stating how they hope to solve the problem. Their purpose and what this manuscript adds to the current literature base are very clear. In general, there is scientific merit for this manuscript. However the following general areas of the paper should be addressed:
- Missing citations – there are several missing citations within the text primarily within the introduction and discussion, please add in the appropriate citations to these areas
- Inconsistency of reporting 2 weeks and 6 months – throughout the entire manuscript there are several versions of how the author reports 2 weeks and 6 months. Some spell the numbers out other places contain the number form, the author should be sure to read the paper for consistency and adjust accordingly.
Specific Section Comments
Abstract
- Within the abstract when reporting the following information, “Participants with less than 7,000 steps per day demonstrated the greatest increase in average daily steps,” it would be helpful if the author reported the actual average change in step count, as well as if this finding was significant or not.
Introduction
- Within paragraph 1, second sentence – this sentence is a little unclear and almost seems like it may be missing a word or two, please provide some clarity to this sentence.
- Within paragraph 1 – please define PA, this is the first time you are using this abbreviation, thus it should be properly defined.
- Within paragraph 1, fourth sentence – can the author clarify what “other ethnicities” are included in the prevalence of physical inactivity reported as 15%.
- Within paragraph 3, second sentence – can the author clarify the threshold of physical activity that is being referred to?
- Within paragraph 4, fourth sentence – the sentence is a little unclear or counterintuitive it states the percent of sedentary behavior and then states 74.6% of adults get any physical activity. Can the author confirm this is what is meant to be taken away from this sentence or adjust if necessary?
- As mentioned above there are a few missing citations throughout the introduction.
Materials and Methods
- Within the statistical analysis section can the author clarify what type of regression analyses were conducted to predict key demographic characteristics of 2-week and 6-month step count averages? A logistic regression is mentioned to determine the odds of baseline step thresholds based on demographic characteristics, but the reviewer is unclear if this is the only type of regression analyses conducted.
Discussion
- As mentioned above there are a few missing citations throughout the discussion.
Tables
- The tables included are very inconsistent. Please make sure you are reporting 2 weeks and 6 months in the same way throughout the tables.
- Table 1:
- The variable age does not have a unit next to it.
- The alignment of the table information is off for variable names and headings.
- Table 2:
- The alignment of the table information is off for variable names.
- Further, the information the authors report as the primary results (i.e. R2) and the sample size are listed at the bottom of the table, which makes it challenging for the reader to find the values. The author should adjust the table with the important information closer to the top to make it easier on the reader.
- Table 3:
- This table alignment is off and the lines/borders do not match up.
- The author only displays the odds ratios for 2 of the 3 step count categories, and the step count categories should have a header over them to clarify what the categories are referring too.
- The variable names have information such as “mean + s.d.” included which is not how the data are reported in this table. The author should adjust the variable names and clearly label what values are being reported.
- Table 4:
- This is the only table that has lines separating each variable, the tables should look very similar even though they are reporting slightly different information.
- The heading “95% C.I.” should be bolded similar to the remaining headings of the table.
